# Nanomaterial Delivery Systems for mRNA Vaccines

**DOI:** 10.3390/vaccines9010065

**Published:** 2021-01-19

**Authors:** Michael D. Buschmann, Manuel J. Carrasco, Suman Alishetty, Mikell Paige, Mohamad Gabriel Alameh, Drew Weissman

**Affiliations:** 1Department of Bioengineering, George Mason University, 4400 University Drive, MS 1J7, Fairfax, VA 22030, USA; mcarras@masonlive.gmu.edu (M.J.C.); salishet@GMU.EDU (S.A.); 2Department of Chemistry & Biochemistry, George Mason University, 4400 University Drive, Fairfax, VA 22030, USA; mpaige3@gmu.edu; 3Perelman School of Medicine, University of Pennsylvania, 130 Stemmler Hall, 3450 Hamilton Walk, Philadelphia, PA 19104, USA; Mg.Alameh@pennmedicine.upenn.edu; 4Perelman School of Medicine, University of Pennsylvania, 410B Hill Pavilion, 380 S. University Ave, Philadelphia, PA 19104, USA; dreww@pennmedicine.upenn.edu

**Keywords:** mRNA, lipid nanoparticle, ionizable lipid, vaccine, SARS-CoV-2

## Abstract

The recent success of mRNA vaccines in SARS-CoV-2 clinical trials is in part due to the development of lipid nanoparticle delivery systems that not only efficiently express the mRNA-encoded immunogen after intramuscular injection, but also play roles as adjuvants and in vaccine reactogenicity. We present an overview of mRNA delivery systems and then focus on the lipid nanoparticles used in the current SARS-CoV-2 vaccine clinical trials. The review concludes with an analysis of the determinants of the performance of lipid nanoparticles in mRNA vaccines.

## 1. Introduction

mRNA vaccines have been propelled to the center stage of the biotechnology and pharmaceutical industry by the COVID-19 pandemic. There are eight ongoing human trials for mRNA vaccines led by BioNTech/Pfizer, Moderna, CureVac, Sanofi/TranslateBio, Arcturus/Duke-NUS Medical School in Singapore, Imperial College London, Chulalongkorn University in Thailand, and Providence Therapeutics [1]. Remarkably, two of these trials have announced interim phase 3 trial results that report an efficacy providing a greater than 94% reduction in SARS-CoV-2 infection after 2 doses of 30 µg or 100 µg of an mRNA sequence encoding for a spike protein immunogen, delivered in a lipid nanoparticle [2,3]. The rapidity of vaccine development also exceeded expectations, with these results occurring only 10 months after the SARS-CoV-2 sequence was made publicly available. This success is a testament not only to the ability of the biotech and pharmaceutical industry to respond to an urgent and unmet global need, but also to the inherent capabilities of mRNA as a pharmaceutical modality, in this case a prophylactic vaccine. The purpose of this review is to overview the development of delivery systems for mRNA and then to summarize the preclinical and clinical findings of the SARS-CoV-2 mRNA vaccines and relate them to characteristics of the delivery system that contribute to their success. Several excellent reviews of mRNA delivery systems for vaccines and therapeutics that predate COVID-19 have been recently published [4,5,6,7,8,9,10,11,12,13,14,15,16].

Messenger RNA therapeutics have many advantages and several challenges compared to other pharmaceutical modalities, including small molecules, DNA, oligonucleotides, viral systems and proteins, including antibodies. The ability to mediate both stimulatory and inhibitory modes of action compared to oligonucleotides and most small molecule drug targets, and to express or replace defective proteins, expands the scope of potential indications for their use. Compared to DNA, mRNA only needs access to the cytoplasmic ribosomal translation machinery rather than the nucleus and does not risk genomic integration. Compared to both proteins and viral systems, mRNA manufacturing is cell-free, faster, and the protein product bears native glycosylation and conformational properties. When combined with a lipid nanoparticle (LNP) delivery system, the nanostructural properties of the mRNA LNP also bear a resemblance to viral systems and circulating endogenous, lipid-containing chylomicrons in terms of their size, lipid envelope and, for viral systems, the internal genomic material that contributes to their application as delivery vehicles for vaccines and other therapeutics [17].

The challenges inherent to the mRNA platform are its intrinsic immunogenicity, susceptibility to enzymatic degradation, and almost negligible levels of cell uptake of naked mRNA. The innate immunogenicity of mRNA is due to the cellular detection of single- and double-stranded RNA by toll like receptors (TLRs)), helicase receptors, including retinoic acid-inducible gene I (RIG-I)-like receptors (RLRs), and others [18,19], which then signal through NF-κB and interferon (IFN) regulatory factors IRF3 and IRF7, which translocate to the nucleus to bind to the type I IFN gene promoter, inducing expression of type I IFNs (IFN-α and IFN-β), accompanied by proinflammatory cytokines, such as tumor necrosis factor-a (TNF-α), IL-6 and IL-12 [20]. The secreted interferons signal through their receptors and the JAK/STAT pathway in the same cell and adjacent cells to activate more than 300 IFN-stimulated genes, including the protein kinase PKR, as a general viral defense mechanism. Although this activation could be beneficial for mounting an immune response to mRNA vaccines, one immediate effect is the downregulation of translation through PKR phosphorylation of eIF2a, which impairs eIF2 activity, inhibiting mRNA translation and thus the protein synthesis of the immunogen [21]. The primary means of abrogating this innate immune response is by substituting naturally occurring nucleosides such as 1-methylpseudouridine [22] and other nucleosides present in transfer and ribosomal RNA (but not typically in mRNA) into the mRNA sequence, which then renders it undetectable via these innate immune sensors [23,24]. This nucleoside-modified immunosilencing mRNA platform is the basis of the mRNA technologies that have recently shown >94% efficacy in the BioNTech/Pfizer and Moderna SARS-CoV-2 vaccine trials, building upon previous trials for other pathogens, which are described in detail below. A second approach pursued by CureVac is sequence engineering involving codon optimization and uridine depletion [25] since TLR7 and TLR8 primarily recognize GU-rich single-stranded RNA sequences [26]. The second challenge for mRNA therapeutics is its susceptibility to nucleases, exemplified by a half-life in serum <5 min [27]. Although chemical modifications of siRNA are highly successful in improving stability and lowering immunogenicity [28], to date, they have not been successful for mRNA due to the sensitivity of the translation machinery to these modifications [29]. The third challenge for mRNA is the lack of cell uptake of naked mRNA in most cell types [30], with the exception of immature dendritic cells [31]. These last two challenges are addressed by the incorporation of a nucleoside-modified or sequence-engineered mRNA into a delivery system that both protects the mRNA from enzymatic attack and facilitates cellular uptake. For example, incorporation into lipid nanoparticles protects the mRNA from enzymatic attack and enhances cell uptake and expression by up to 1000-fold compared to naked mRNA when administered in animal models [32,33].

Therapeutic mRNA is produced by in vitro transcription (IVT) from a plasmid DNA backbone to produce a full length message bearing a 5′ cap, a 5′ untranslated sequence (UTR), the open reading frame coding for the protein of interest, the 3′UTR and a polyA tail [4]. The natural eukaryotic 5′ cap (cap0) is an inverted 7-methyl guanosine (m7G) linked to the first nucleotide of the mRNA by a 5′ to 5′ triphosphate. Cap0 protects endogenous mRNA from nuclease attack, is involved in nuclear export and binds to translation initiation factor 4 to start protein translation. Two additional 5′ caps have been identified (cap1 and cap2) that contain additional methyl groups on the second or third ribonucleotide and are less immunogenic than cap0 (and therefore preferred) [34]. A commonly used current capping method involves a co-transcriptional capping process resulting in cap1, which possesses high translation and low immunogenicity [35]. The 5′UTR is involved in translation initiation and can contain a Kozak sequence as well as an internal ribosomal entry site for cap-independent translation [36]. The open reading frame is followed by the 3′UTR, which influences mRNA stability and durability of protein expression. The polyA tail is encoded at around 100 residues and helps initiate translation and delay degradation. IVT production of mRNA needs to be followed by careful purification to remove DNA and double-stranded RNA contaminants, which are immunogenic [37,38]. The mRNAs described above can be nucleoside modified or sequence engineered without nucleoside modification, but are not capable of self-replication. Self-amplifying mRNA (samRNA) capable of replication are also being tested in clinical trials for SARS-CoV-2 and are longer ~10 kb sequences since they contain four additionally encoded nonstructural genes, including an RNA-dependent RNA polymerase, which result in self-replication inside cells but do not produce an infectious particle since they lack structural genes [39]. samRNAs cannot be nucleoside modified since these modifications interfere with self-amplification. Due to the amplification process, samRNAs typically use lower doses (1–10 µg) in the current COVID-19 clinical trials compared to 30–100 µg for non-amplifying mRNA. Interestingly, all of the above categories of mRNA vaccines are currently being tested in human clinical trials for SARS-CoV-2 and are summarized in Table 1. All mRNA delivery systems in these clinical trials are lipid nanoparticles. The exact composition of the Pfizer-BioNTech LNP [40] and Moderna LNP [41] have been publicly disclosed, while some others have not. The others are all most likely similar to the Alnylam Onpattro™ product (described further below) but with a proprietary ionizable lipid, as is the case for those that are disclosed. Although the specific ionizable lipid used may not be known in all cases, its general class can be understood from journal and patent publications and is indicated in Table 1.

Prior to COVID-19, mRNA vaccines were used in preclinical and clinical studies for infectious diseases including influenza, zika, HIV, Ebola, rabies, chikungunya, malaria, genital herpes, toxoplasma gondii, and others. These studies are summarized in a number of excellent recent reviews [4,6,16,39].

## 2. Early Delivery Systems for mRNA Vaccines

Protamine, a mixture of small arginine-rich cationic proteins, has been used to form complexes with mRNA that improved transfection compared to naked mRNA [62]. Later, a mixture of free mRNA with protamine-complexed mRNA was introduced [63] since protamine-complexed mRNA partly inhibited protein expression [64]. Dynamic light scattering indicated that free mRNA have a size near 50 nm, while the protamine/mRNA complexes were in the 250–350 nm range [63]. This approach was pursued by CureVac for a rabies vaccine candidate, CV7201, a lyophilized, temperature-stable non-modified mRNA composed of free and protamine-complexed mRNA encoding the rabies virus glycoprotein (RABV-G) [65]. In Balb/c mice, two doses of 10 µg and higher induced neutralizing titers greater than the WHO threshold of protection and administration of an 80 µg dose twice was protective against a lethal intracerebral challenge [66]. In a phase 1 human trial using doses 80–640 µg applied through intradermal and intramuscular routes, only a subgroup of participants who received three 80–400 ug doses using a particular injector device achieved the WHO neutralization titer threshold [67]. A serious adverse event (Bell’s Palsy) occurred for one participant out of 101 at the highest dose and 5% of all participants experienced a solicited severe adverse event. The overall rate of all adverse events was high, with 97% experiencing injection site reactions and 78% a systemic adverse event. Given this suboptimal delivery with protamine complexed mRNA, CureVac adopted a lipid nanoparticle delivery system from Acuitas [47,68] and demonstrated greatly improved neutralizing titers at a 20-fold lower dose of 0.5 µg (vs. 10 µg for protamine complexed mRNA) in Balb/c mice and at a 10 µg dose in non-human primates [69]. Activation of T cell responses and the presence of IL-6 and TNF in the draining lymph nodes and injection sites indicated the role of the LNP in mediating the positive immune response. A clinical trial has been initiated (NCT03713086), with interim results expected to be reported in 2021.

A cationic nanoemulsion (CNE) was developed for mRNA delivery by combining the cationic lipid DOTAP with a commercial adjuvant (MF59) containing squalene, sorbitan trioleate, and polysorbate 80 in a citrate buffer of pH 6.5 [70]. The combined use of a self-amplifying mRNA encoding for respiratory syncytial virus glycoprotein (RSV-f) with an NP amine (from DOTAP) to phosphate ratio (of mRNA) of 7 resulted in an average 129-nm sized nanoparticle. One advantage of this approach is the ability to store CNE and mRNA separately and combine them only at the time of use. A 15-µg dose administered twice in Balb/c mice elicited neutralizing titers above that of an adjuvanted subunit vaccine. Detectable neutralization titers and T cell responses in non-human primates were achieved with two doses of 75 µg. Building on this concept, a separate group created a Nanostructured Lipid Carrier (NLC), which is a hybrid between a CNE and a lipid nanoparticle, consisting of a liquid oil phase, such as squalene, with a solid-phase lipid composed of a saturated triglyceride [71]. NLCs containing a self-amplifying mRNA encoding for a sika immunogen had a particle size of 40 nm and an NP ratio of 15 and were capable of generating protective neutralizing titers in C57BL/6 mice after a single injection of a dose as low as 0.1 µg or 0.01 µg.

## 3. Polymers for mRNA Delivery

Cationic polymers have been widely used for nucleic acid delivery for several decades, including for example poly(L-lysine), polyethylenimine (PEI), DEAE-dextran, poly(β-amino esters) (PBAE) and chitosan. In their simplest format, cationic polymers are mixed in excess with nucleic acid to form electrostatically bound cationic polyplexes. Although many polymers have been developed, they are not as advanced as lipid nanoparticles for nucleic acid delivery and the number of animal studies applying them successfully to vaccines is limited. PBAEs were co-formulated with polyethylene glycol (PEG)-lipids to form mRNA/PBAE/PEG–lipid nanoparticles that were capable of the functional delivery of mRNA to the lungs after intravenous administration in mice [72]. A biodegradable polymer, poly(amine-co-ester) (PACE) terpolymer, has been examined for mRNA delivery using erythropoietin as a reporter post-IV administration for gene delivery [73]. By controlling the molecular weight and end group chemistry, a 10 kDa member of the PACE family achieved the same in vitro transfection efficiency as TransIT, a potent but toxic colloidally unstable and large-sized commercial reference. In vivo expression of EPO at 20 µg IV was fivefold more potent than TransIT. Hyperbranched poly (beta amino esters) (hPBAEs) were synthesized for mRNA delivery to the lung by inhalation. hPBAE mRNA polyplexes were 137 nm in size and were able to transfect 25% of the lung endothelium when nebulized and inhaled in mice without evident toxicity and with expression levels 10-fold that of branched PEI [74]. A disulfide-linked poly(amido amine), pABOL, was synthesized at molecular weights ranging from 8 kDa to 167 kDa and was able to form polydisperse nanocomplexes near 100 nm in size [75]. In vivo luciferase expression of these polyplexes using a self-amplifying mRNA reporter was similar to that of PEI after intramuscular administration. When delivered to mice with a hemagglutinin (HA) influenza immunogen in a prime-boost design, neutralizing titers were highest for the low molecular weight 8 kDa pABOL and exceeded those of PEI. The 8 kDa pABOL delivering 1 µg HA of self-amplifying mRNA was also partly protective against a lethal influenza challenge, preventing death but not preventing significant weight loss. This pABOL system was considered for the delivery of a self-amplifying mRNA immunogen for SARS-CoV-2 by the group at Imperial College London; however, delivery of a SARS-CoV-2 immunogen with pABOL was 1000X less potent than delivery of the same immunogen with an optimized lipid nanoparticle from Acuitas [59]. In total, 1µg of self-amplifying RNA in pABOL generated the same binding antibody and neutralization titers as 0.001 µg in an optimized lipid nanoparticle (Dr. Anna Blakney, personal communication). Many other polymer systems are capable of delivering mRNA in vitro or in vivo but remain to be tested in a vaccine context [76,77,78,79,80,81,82,83,84].

## 4. Development of Lipid Nanoparticles for the Current SARS-CoV-2 Clinical Trials

The earliest transfection reagent for mRNA was the quaternized cationic DOTAP combined with ionizable and fusogenic DOPE, adopted from DNA transfection [85] for the transfection of mRNA in numerous cell types [86]. Although effective in vitro, the permanently cationic quaternized ammonium group renders these large-sized lipoplexes rapidly cleared from circulation and from generally targeting lungs, as well as exhibiting toxicity. The forerunner of today’s LNP was the stabilized plasmid–lipid particle (SPLP) that was formed by combining the fusogenic ionizable DOPE with a quaternized cationic lipid, DODAC, which electrostatically bound and encapsulated plasmid DNA, which was then coated with hydrophilic PEG to stabilize it in aqueous media and limit protein and cell interactions upon administration in vivo [87]. DOPE can be protonated in the endosome after cell uptake and, since it is cone-shaped, it can form an endosomolytic ion pair with endosomal phospholipids to facilitate endosomal release, a critical event for successful delivery [17]. The SPLP was then further developed as a Stabilized Nucleic Acid Lipid Particle (SNALP) containing siRNA that included four lipids: an ionizable rather than quaternized cationic lipid, a saturated bilayer forming quaternized zwitterionic lipid, DSPC, cholesterol and a PEG–lipid [88]. In addition to electrostatically binding to the nucleic acid, the ionizable lipid in the SNALPs played the role of the fusogenic lipid and became protonated in the endosome to form a membrane-destabilizing ion pair with an endosomal phospholipid. It is now known that DSPC helps form a stable bilayer underneath the PEG surface [89]. Cholesterol plays several roles, including filling gaps in the particle, limiting LNP–protein interactions and possibly promoting membrane fusion [90]. The ionizable lipid plays a central role by being neutral at physiological pH, thus eliminating any cationic charge in circulation, but becoming protonated in the endosome at pH ~6.5 to facilitate endosomal release. The development of the first siRNA product that was clinically approved in 2018 primarily focused on optimizing the ionizable lipid and, secondarily, the PEG–lipid and the ratios of the four lipids used in the LNP, as well as the LNP assembly and manufacturing procedure. An optimal number of unsaturated bonds in the C18 tail were found to be providing a dilinoleic acid tail linked by ethers to a dimethylamine headgroup [88], in accordance with the molecular shape hypothesis [12,91]. However, the introduction of a single linker to the dilinoleic acid tail, which had an optimized number of carbons from the dimethylamine head group to the linker, resulted in the pKa of the ionizable lipid in the LNP being near 6.4 for the ionizable lipid DLin-MC3-DMA [92,93]. The last step in the optimization was to tune the mole ratios of these lipids to 50/10/38.5/1.5 for MC3/DSPC/Cholesterol/PEG–lipid. Overall, this optimization process from DLin-DMA to DLin-MC3-DMA required more than 300 ionizable lipids to be screened in thousands of formulations and resulted in a 200-fold increase in potency and a corresponding reduction in the effective dose in order to achieve durable suppression of the target gene >80% and a therapeutic window that permitted the clinical approval of Onpattro™ in 2018 [94,95]. This MC3 formulation developed for siRNA is the basis for the subsequent development of LNPs described below (Figure 1), which are now under emergency use after being approved for the delivery of SARS-CoV-2 mRNA vaccines.

Moderna carried out several preclinical [97,98,99] and clinical studies [97,100] using MC3 in the Onpattro formulation described above in order to deliver nucleoside-modified mRNA-encoded immunogens. MC3 was later identified [42,101] as the ionizable lipid in these studies comparing a new class of ionizable lipids to MC3. This new class includes Lipid H [42], which is the ionizable lipid SM-102 [41] in Moderna’s SARS-CoV-2 product mRNA-1273 (Table 2). Using a nucleoside-modified mRNA-encoded immunogen for the Zika virus, the MC3 LNP was capable of protecting immunocompromised mice lacking type I and II interferon (IFN) signaling against a lethal challenge with one 10 µg dose or two 2 µg doses in a prime-boost design [99]. Similar results were obtained in immunocompetent mice pre-administered with an anti-ifnar1 blocking antibody to create a lethal model. In a series of influenza studies delivering nucleoside-modified mRNA-encoding hemagglutinin (HA) immunogens, the MC3 LNP delivered intradermally was capable of fully protecting mice against a lethal challenge with a single dose as low 0.4 µg, although post-challenge weight loss occurred even when up to 10 µg of a single dose was administered [97]. A single dose of 50 µg or 100 µg produced high HAI (hemagglutination inhibition assay) titers in ferrets, as did two doses of 200 or 400 µg in non-human primates. In a small number (23) of human subjects who received 100 µg doses, all had HAI titers >40 (the WHO correlate of protection) that were more than fourfold above the baseline at the beginning of the study. In a larger phase 1 trial using these same MC3 LNPs delivering two distinct nucleoside-modified mRNA-encoded HA immunogens, intramuscular injection of 100 µg of the H10N8 immunogen resulted in 100% of the 23 subjects having HAI titers >40 [100]. Although no life-threatening adverse events occurred, 3 of these 23 subjects experienced severe grade 3 adverse events. A planned 400 µg dose was discontinued after two of three subjects experienced grade 3 adverse events, which met the study pause rules. At lower doses, the frequency and severity of adverse events diminished, although nearly every subject experienced at least one adverse event. These studies were promising, but also highlighted the relatively narrow therapeutic window to obtain protective immunizations at doses that do not cause a problematic number of adverse events. This is reminiscent of the narrow therapeutic window of the MC3 precursor, DLin-DMA, which needed an improved potency in order to lower the dose and still achieve efficacious gene knockdown.

Since siRNA products require repeated dosing for chronic diseases, there was a concern that the slow degradability of the dilinoleic alkyl tail in MC3 would cause accumulation and potential toxicity with repeated dosing. A biodegradable version of MC3, Lipid 319 (Table 2), was generated by replacing one of the two double bonds in each alkyl chain with a primary ester that can be easily degraded by esterases in vivo [68]. A half-life of less than an hour in the liver was noted for Lipid 319, while it maintained a gene silencing efficiency in the liver that was similar to MC3. The degradation products were confirmed in vivo, as well as their secretion and the nontoxic nature of Lipid 319. This study of Lipid 319 is cited in the preclinical and clinical studies for SARS-CoV-2 as representing the Acuitas LNP class used in the BioNTech [49] and CureVac [53,69] products, although the Acuitas LNP delivering the self-amplifying RNA in the Imperial College London trial [60] is cited as having been contained in a more recent patent application [59], represented here by Lipid A9 from Acuitas (Table 2). Recently, the identity of the Acuitas ionizable lipid in BioNTech’s approved BNT162b2 was disclosed [40] as ALC-0315 (Table 2). An important aspect of these LNPs is that they were developed by screening mRNA expression in the liver following IV administration and may not yet be fully optimized for the intramuscular administration of mRNA-based vaccines.

Moderna recently developed a new class of ionizable lipids to replace MC3, primarily due to the above-mentioned concerns related to the slow degradability of MC3, but also with the effort of increasing their potency by enabling greater branching than the dilinoleic MC3 alkyl tail [42,101]. This new class of lipids has an ethanolamine ionizable head group, connected to both a single saturated tail containing a primary degradable ester—like that of Maier 2013—and a second saturated tail that branches after seven carbons into two saturated C8 tails using a less degradable secondary ester, as in Lipid 5 [101] (Table 2), optimized for IV administration to the liver, and a similar Lipid H [42] or SM-102, found to be optimal for the intramuscular (IM) administration of vaccines. Increased branching is a common feature pursued by Acuitas, as Lipid A9 has a total of five branched chains [59] (Table 2) vs. three for the Moderna LNPs. Increased branching is believed to create an ionizable lipid with a more cone-shaped structure, so that—when paired with the anionic phospholipid in the endosome—a greater membrane-disrupting ability will occur, following the molecular shape hypothesis outlined several decades ago [12,91]. When administered IV, Lipid 5 was not detectable in the liver at 24 h, while MC3 was present in the liver at 71% of its initial dose, verifying the degradability of Lipid 5. Lipid 5 was three-fold more potent than MC3 in mice for luciferase expression and five-fold more potent in non-human primates for hEPO after IV administration. These increases in potency were consistent with and possibly caused by an increase in endosomal release, with up to 15% of the mRNA in the cell being released from the endosome for Lipid 5 versus 2.5% for MC3, the latter being similar to that previously measured for MC3 using siRNA [106]. However, cell uptake in these endosomal release experiments was fourfold higher for MC3 vs. Lipid 5 so that absolute amounts of released mRNA in the cytoplasm were similar for these two LNPs. The same ionizable lipid library was examined in intramuscular administration for vaccines and was similarly found to be degradable and quickly eliminated due to the primary ester and generally to have a 3–6 fold increase in potency in terms of protein expression or immunogenicity compared to MC3 for an influenza nucleoside-modified mRNA encoded immunogen in mice, although immunogenicity in non-human primates was identical to MC3 at a 5 µg prime-boost dose [42]. Lipid H or SM-102 (Table 2) was identified as the optimal candidate and structurally only differs from Lipid 5, identified as optimal for IV administration, by a two-carbon displacement of the primary ester. The pKa of Lipid 5 LNP was 6.56, while that of the Lipid H LNP was 6.68, suggesting that a slight increase in pKa may be beneficial for IM vs. IV administration, although this difference is within the variability of the assay. Histological examination of muscle injection sites in rats indicated that Lipid H LNPs attracted less of a neutrophil- and macrophage-enriched inflammatory infiltrate compared to MC3, which may reduce injection site reactogenicity in human trials [42].

## 5. mRNA Lipid Nanoparticles in the Current SARS-CoV-2 Clinical Trials

### 5.1. BioNTech/Pfizer

Acuitas ALC-0315 (Table 2) combined with DSPC, cholesterol and a PEG–lipid is the delivery system in the SARS-COV-2 trials of BioNTech [40]. CureVac and Imperial College London may also use ALC-0315, or possibly A9 (Table 2). BioNTech began developing its SARS-CoV-2 vaccine with four mRNA-encoded immunogens, two of which were nucleoside modified, one unmodified and one self-amplifying. Reports are available for the two nucleoside-modified mRNAs: BNT162b1 is a short ~1 kb sequence encoding the receptor-binding domain of the spike protein, modified by a foldon trimerization domain to increase immunogenicity by multivalent display. The longer 4.3 kb BNT162b2 encodes a diproline-stabilized, full-length, membrane-bound spike protein. BNT162b2 received EU and US emergency approval recently. In a preclinical study, binding antibodies and neutralization titers in mice were detectable after a single dose of 0.2, 1, and 5µg of BNT162b2, increasing by an order of magnitude from the lowest to the highest dose and eliciting strong antigen-specific Th1 IFNγ and IL-2 responses in CD4+ and CD8+ splenocytes with very low levels of Th2 cytokines [49]. Draining lymph nodes also contained high numbers of germinal center B cells and elevated counts of CD4+ and CD8+ T follicular helper (Tfh) cells, which were previously identified as partly induced by the LNP alone in mRNA LNP vaccines [33]. In non-human primates, prime-boost doses of either 30 µg or 100 µg elicited binding antibody and neutralization titers that were more than 10 fold those of a human convalescent panel and a strongly Th1-biased T cell response that is believed to be important to protect against vaccine-associated enhanced respiratory disease [107]. In a limited number (6) of challenged rhesus macaques, two doses of 100 µg rendered undetectable viral titers in bronchoalveolar lavage and from nasal swabs. A phase 1 clinical trial for the smaller mRNA-encoded immunogen BNT162b1 planned 10, 30 and 100 µg doses on day 1 and day 21. The intermediate dose of 30 µg induced antibody binding and neutralization titers that were 30-fold and threefold higher than those of a human convalescent panel, respectively. The 100 µg dose was not administered for the boost due to the presence of severe injection site pain after the first dose. Injection site pain was reported by 100% of subjects with the 30 µg boost, but at mild or moderate severity. Following the second vaccination at the 30 µg dose, nearly all subjects experienced mild or moderate systemic adverse events of fever, chills or fatigue. This trial also demonstrated strong Th1-biased T cell responses from peripheral blood mononuclear cells [50]. A phase 2 trial compared both BNT162b1 and BNT162b2 in groups of younger (18–55 y) and older (65–85 y) subjects [51]. Binding and neutralizing antibody titers were slightly lower in the older subjects, but still exceeded those in a convalescent panel. The severity of adverse reactions was also reduced in the older versus younger subjects. A significant reduction by ~twofold in the frequency of systemic adverse events (fever, chills, fatigue) was found in BNT162b2 versus BNT162b1. It was this increase in the tolerability of BNT162b2 that drove its selection for the phase 3 trial, where a 94% effectiveness was recently announced, since 162 COVID-19 cases occurred in the placebo arm, while only 8 cases were found in the vaccinated group that received two 30 µg doses of BNT162b2 [3].

### 5.2. Moderna

The nucleoside-modified mRNA encoded immunogen in Moderna’s studies is a transmembrane-anchored diproline-stabilized prefusion spike with a native furin cleavage site and is delivered in an LNP that follows the prototype MC3 LNP, but replaces MC3 with Lipid H (SM-102) [41,42]. This mRNA LNP (mRNA-1273) induced neutralizing antibodies in several mouse species when injected at 1 and 21 days with a 1µg dose, but not at a 0.1 µg dose [44]. The T cell response appeared to be a balanced Th1/Th2 response and viral titers in mice lungs and nasal turbinates in a mouse-adapted virus challenge model were reduced to baseline with two doses of 1 µg, but not with 0.1 µg. In rhesus macaques, 2 doses of 100 µg produced high binding and neutralizing titers and a Th1-biased response in peripheral blood that also involved a strong Tfh response [45]. Titers and T cell responses were significantly lower with two 10 µg doses. Similarly, the 100 µg dose was capable of reducing viral titers in bronchoalveolar lavages and nasal swabs to baseline, while 10 µg only did so in the lungs. In a phase 1 study with 15 patients per group receiving 2 doses of 25, 100 or 250 µg, separated by 4 weeks, binding and neutralization titers were ~10-fold higher than convalescent for the 100 µg dose, and about equivalent to convalescent at 25 µg [46]. Solicited adverse events were report by all subjects at the 100 µg and 250 µg doses and 3 of 14 in the 250 µg group reported severe adverse events and were discontinued. In a subsequent phase 1 study in older patients (56–71 y and above 71 y), the 25 µg and 100 µg doses were found to produce binding antibody titers above those of convalescent plasma, while neutralizing titers were equivalent at 100 µg, but lower than convalescent at 25 µg [43]. Most patients (~80%) still experienced adverse events after the second vaccination, even in the older age group. Analyses of peripheral blood showed a CD4 T cell response that was Th1 biased. The higher neutralization titers for the 100 µg dose vs. the 25 µg dose resulted in its selection for the phase 3 trial, where interim results announced a 94.5% efficacy with 90 cases of COVID-19 in the placebo group versus five in the vaccinated group [2]. An independent board conducted an interim analysis of Moderna’s phase 3 trial and found that severe adverse events included fatigue in 9.7% of participants, muscle pain in 8.9%, joint pain in 5.2%, and headache in 4.5%, while, in the Pfizer/BioNTech phase 3 trial, the frequency was lower with fatigue at 3.8% and headache 2% [108].

### 5.3. CureVac

The CureVac mRNA LNP (CVnCoV) is a non-chemically modified, sequence-engineered mRNA encoding a diproline stabilized full-length S protein delivered in an Acuitas LNP, possibly using the ionizable lipid ALC-0315. The number of weeks between two doses was examined ranging, from 1 to 4 when using 2 µg doses in mice, where it was found that the longer intervals produced higher titers and T cell responses and a balanced Th1/Th2 response in Balb/c mice [53]. The second dose was required to produce neutralizing antibodies and two doses of 0.25 µg were insufficient to produce neutralizing antibodies. In Syrian golden hamsters, two 10 µg doses (but not 2 µg) were able to reduce viral titers in the lungs (but not nasal turbinates) to baseline. In a phase 1 clinical trial examining 2–12 µg doses, neutralizing titers reaching the levels of convalescent sera were only found at the highest 12 µg dose, resulting in higher doses of 16 and 20 µg being included in the ongoing phase 2 trial [52]. All patients at the 12 µg dose experienced systemic adverse events after each dose, the majority being moderate and severe, while >80% experienced local injection site pain at the mild and moderate levels.

### 5.4. TranslateBio

Translate Bio uses a non-modified mRNA encoding a double mutant form of the diproline stabilized spike protein delivered in an LNP that is cited as being based on the ionizable lipid C12-200 [109], but may be a more recently synthesized candidate from the ICE- [110] or cysteine-based [55] ionizable lipid families. In Balb/c mice, two doses in the range of 0.2–10 µg resulted in binding and neutralization titers well above convalescent levels. In non-human primates 15, 45 and 135 µg doses all generated titers exceeding the human convalescent panel [56]. The immune response was also Th1 biased.

### 5.5. Arcturus

Arcturus uses a self-amplifying, full-length, unmodified mRNA encoding a pre-fusion SARS-CoV-2 full-length spike protein in an LNP that uses an ionizable lipid with a thioester to link the amine-bearing headgroup to lipid tails via two additional ester groups. Two possible ionizable lipids in this family are Lipid 10a (in Table 4 of [111]) or Lipid 2,2 (8,8) 4C CH3 (on p. 33 of [57]) (Table 2). The latter has three branches, resembling the Moderna Lipid H, but with a degradable thioester linked to the headgroup. A feature of self-amplifying mRNA was observed where luciferase reporter expression was maintained at a fairly constant level beyond one week of IM administration, while conventional mRNA expression fell quickly [58]. The vaccination alone surprisingly produced weight loss and increased clinical scores in C57BL/6 mice. Only a single dose at 2 µg or 10 µg (but not 0.2 µg) in mice was required to reach neutralization titers above 100 in a Th1-biased response with high levels of antigen-specific T cell responses. A single dose of 2 µg or 10 µg was also 100% protective in the K18-hACE2 lethal mouse challenge model, generating 100% survival with no weight loss and a reduction in lung and brain viral titers to baseline. Arcturus has completed a phase 1 clinical trial with doses from 1–10 µg and has chosen 7.5 µg for its phase 3 trial [112].

### 5.6. Imperial College London

Imperial College London uses a self-amplifying mRNA-encoded prefusion-stabilized spike protein delivered in an Acuitas LNP, which is described in the patent [59] represented by Lipid A9 [60] (Table 2). Remarkably high and dose-dependent antibody and neutralizing titers were obtained after two injections of doses in the range 0.01 µg to 10 µg in Balb/c mice. The response was strongly Th1 biased and the 10 and 1 µg doses produced threefold higher antigen-specific splenocyte responses compared to the lower 0.1 and 0.01 µg doses. A phase 1 clinical trial is about to start for this vaccine.

### 5.7. Chulalongkorn University, University of Pennsylvania

Chulalongkorn University, in collaboration with the University of Pennsylvania, is developing a native spike immunogen nucleoside-modified mRNA LNP using a Genevant LNP, likely CL1 Lipid [61]. They aim to begin phase 1 clinical trials in Q1 of 2021 and begin distribution of the vaccine in Q4 of 2021 to Thailand and seven surrounding low to moderate income countries.

### 5.8. Providence Therapeutics

Providence therapeutics was granted a Health Canada notice of authorization to pursue human clinical trials for the PTX-COVID-19B mRNA LNP vaccine [113]. Preclinical studies of three mRNA candidates encoding the receptor-binding domain, the full-length spike with or without a mutation in the furin cleavage site, were administered at a dose of 20 µg in C57BL6 mice following a prime-boost regimen [114]. Preclinical data using an undisclosed lipid from Genevant, possibly similar to CL1 In Table 2, showed robust neutralization titers for the full length and the furin-mutated payloads, reminiscent of the data observed in [115]. Phase 1 clinical trials are scheduled to begin in Q1 of 2021, with manufacturing and distribution of the vaccine—pending regulatory approval—in the same year.

### 5.9. Storage and Distribution

Most RNA LNPs made in the laboratory are stable at 4 °C for several days, but then exhibit size increases and a gradual loss of bioactivity, such as luciferase expression [116]. A size increase over time from LNP aggregation has been commonly observed in previous siRNA LNP formulations [117]. In order to stabilize mRNA LNP vaccines for storage and distribution, a frozen format has been required to date. The Moderna COVID-19 vaccine needs to be stored from −25 °C to −15 °C, but is also stable between 2 °C and 8 °C for up to 30 days and between 8 °C and 25 °C for up to 12 h [118]. The Pfizer/BioNTech COVID-19 vaccine needs to be stored from −80 °C to −60 °C and then thawed and stored from 2 °C to 8 °C for up to 5 days prior to dilution with saline before injection [119]. The dry ice temperatures required for the Pfizer vaccine are more difficult to achieve during distribution and storage than the regular freezer temperature required by the Moderna vaccine. The reasons behind these temperature differences are not obvious since both vaccines contain similar high concentrations of sucrose as a cryoprotectant. The Moderna mRNA LNPs are frozen in two buffers, Tris and acetate [41], while the Pfizer/BioNTech vaccine only uses a phosphate buffer [40]. Phosphate buffers are known to be suboptimal for freezing due to their propensity to precipitate and cause abrupt pH changes upon the onset of ice crystallization [120,121]. Lyophilization has been challenging for mRNA LNPs [116]. However, Arcturus has stated that their COVID-19 mRNA vaccine is stable in a lyophilized format, which would presumably greatly simplify distribution, although the temperature stability of this lyophilized formulation has not yet been disclosed [122].

## 6. Lipidoid Nanoparticles

A number of lipid-like entities, termed lipidoids, were initially developed for siRNA delivery and subsequently used for mRNA delivery. One example is C12-200 (Table 2), which was selected from a large lipidoid family due to its high efficiency in hepatocyte gene silencing via IV administration [123]. For efficient liver-directed gene silencing, C12-200 was combined with the same lipids as the MC3 Onpattro prototype, namely 50% ionizable lipid, 10% DSPC, 38.5% cholesterol and 1.5% PEG–lipid. A later study found that the C12-200 delivery efficiency for mRNA to the same liver target could be increased sevenfold by reducing the percentage of ionizable lipid to 35%, but increasing the weight ratio of ionizable lipid to nucleic acid from 5 to 10 and replacing DSPC with the fusogenic unsaturated DOPE [103]. Interestingly, this optimized formulation increased mRNA expression sevenfold, but did not change the silencing efficiency for siRNA. C12-200, in this formulation, has also been studied for mRNA-mediated protein replacement therapy in mice and nonhuman primates [124], but was seen to generate a strong inflammatory response by histology when injected subcutaneously [109]. C12-200 is a small molecule dendrimer with five alkyl chains and five nitrogen atoms, three of which appear to be protonatable, according to ionization analyses that can be performed with commercial software such as ACDLabs Percepta (Table 2). Another dendrimer lipidoid, 5A2-SC8, was identified for high siRNA delivery efficiency to the liver in a separate screening process, and also has five nitrogen atoms and five short alkyl chains [125] (Table 2). The 5A2-SC8 lipidoid had poor efficiency for mRNA delivery unless its formulation parameters were similarly changed by lowering the ionizable lipid mole fraction to 24%, using DOPE instead of DSPC, and increasing the other lipid proportions, but, at the same time, increasing the weight ratio of 5A2-SC8 to mRNA to 20 [104]. These formulation changes appear to be needed for these dendrimer-type lipidoids to be effective mRNA delivery vehicles, possibly since they have multiprotic head groups and a dendrimer structure. Another very high molecular weight modified dendrimer was used to deliver self-amplifying mRNA encoding immunogens for influenza, Ebola and toxoplasma gondii and was shown to be protective against all three pathogens in mice after a single, very high dose of 40 µg or prime-boost 4 µg injections, which is also a high dose for replicating RNA [126]. An interesting recent finding for a series of lipidoids was that an additional single carbon branch at the terminus of each of the four alkyl chains of this small, three-nitrogen dendrimer increased the potency of liver expression more than 10-fold compared to other lipidoids in this class [105]. There was no correlation of this increased potency with the LNP pKa, but there was a correlation with the absolute fluorescence of the TNS dye at pH 5, which indicates that the amplitude of protonation in the endosome correlates to mRNA expression, presumably by facilitating endosomal release. The additional carbon branch could also be expected to produce a more cone-shaped structure and thereby more membrane disruption according to the molecular shape hypothesis [12,91].

## 7. Intranasal Delivery of mRNA Lipid Nanoparticles

For mRNA vaccines, the vast majority of studies and all current clinical trials have used intramuscular administration, while intradermal administration has also been studied, usually in parallel with the intramuscular route. Although not highly developed to date, intranasal administration of vaccines presents advantages such as the activation of mucosal immunity, which is very relevant for respiratory pathogens, and a reduced reliance on needle-based immunizations. The MC3 LNP has been used to deliver a 4.5 kb nucleoside-modified sequence encoding the cystic fibrosis transmembrane conductance regulator (CFTR) to mice [127]. A luciferase reporter was successfully expressed in the lungs by pipetting a 12 µg dose into the nostrils for spontaneous inhalation. Then, in a transgenic CFTR knockout mouse, application of CFTR mRNA LNPs restored CFTR-mediated chloride secretion to conductive airway epithelia for at least 14 days. MC3 LNPs were used again in a subsequent study of delivery to the nasal epithelium by using a luciferase reporter. Here, the use of a nebulizer to create an aerosol using the LNPs was examined; however, aerosolization resulted in LNP aggregation, doubling their size to 170 nm and resulting in a loss of transfection activity in vitro [128]. As a result, the researchers decided to instill the LNPs into the nostrils and found the luciferase reporter mainly expressed in nasal epithelia, with some additional transfection in lung epithelia. This study highlighted the delivery challenges of obtaining uniform and high levels of mRNA transfection in nasal and lung epithelia. Intranasal delivery of mRNA LNPs was also achieved using the older DOTAP/cholesterol/PEG–lipid system combined with protamine to encapsulate non-modified mRNA-expressing cytokeratin 19 in order to provoke a cellular immune response and slow tumor growth in a Lewis lung cancer xenograft model in mice [129]. These LNPs were large, 170 nm in size, and cationic, with 10 mV zeta potential and the ability to transfect 30% of DC2.4 dendritic cells in vitro. Once the xenograft tumor was established, 10 µg of cytokeratin 14-encoding mRNA LNPs was intranasally instilled in 100 µL PBS once per week for 3 weeks, resulting in a very significant reduction in tumor volume growth compared to the PBS control. A nucleoside-modified mRNA encoding the influenza antigen H3N2-HA was delivered in another study using DOTAP/DOPE/PEG–lipid LNPs, as well as in the same LNP-bearing mannose as a ligand to facilitate uptake by macrophages and dendritic cells [130]. These LNPs were also large, at 200 nm, positively charged, at 15 mV zeta potential, and able to express luciferase in the lungs following intranasal instillation of a 12 µg dose. Two 12 µg doses of the H3N2-HA LNPs were instilled intranasally at weeks 0 and 3 in C57BL/6 mice that were subsequently challenged with a lethal dose of H1N1. Both LNPs containing the mRNA-encoded immunogen were capable of complete protection, while the mannose-coated LNP appeared more able to also block weight loss. Intranasal administration of LNPs appears feasible, although the doses were higher than those reported for intramuscular administration and the method of installation or aerosolization still requires further development.

## 8. Delivery of mRNA LNPs Encoding Antibodies

More than 70 monoclonal antibodies (mAbs) are currently on the market, with global sales of 125 billion USD. The possibility of using mRNA-encoded antibodies may bring some advantages, including endogenous protein synthesis benefiting from native post-translation modifications and a simplified manufacturing method that does not require cell culture and extensive purification and characterization of the protein product [8]. The feasibility of delivering mRNA-encoded mAbs for passive immunization was shown by the encapsulation of purified nucleoside-modified mRNAs encoding the light and heavy chains of VRC01, a broadly neutralizing antibody against HIV-1, into Acuitas LNPs [131]. Balb/c mice receiving a 30 µg dose IV that would target hepatocytes expressed the mAbs for more than a week, with serum levels reaching 150 µg/mL, which was higher than that obtained by direct injection of 600 µg of the mAb, with weekly injections capable of maintaining a constant serum level above 40 µg/mL. Both a 30 µg and a 15 µg injection of the mRNA LNP could protect CD34-NSG humanized mice from an HIV-1 challenge given 24 h later, as indicated by analyses of serum for viral RNA copies 2 weeks post challenge. The feasibility of therapeutic non-modified mRNA-encoded antibodies was confirmed in a study by CureVac, also using Acuitas LNPs [25], where an IgG mAb with broad neutralization ability for a variety of rabies strains was chosen, as well as a heavy chain-only Vh domain-based (VHH) neutralizing agent against the botulinum toxin [132]. An mRNA-encoded rituximab, targeting CD20, the gold standard for treating non-Hodgkin’s lymphoma, was also produced. The animal studies here used an Acuitas LNP that targeted hepatocytes by IV administration. A single administration of 40 µg in mice produced serum levels of IgG just above 10 µg/mL, which gradually declined to 1 µg/mL after 1 month. The same dose of the VHH single-domain neutralizing agent produced 10-fold higher levels, but with a much shorter half-life of several days due to the absence of the Fc region. Single IV administration of 40 µg in mice was also able to entirely protect mice when administered either 1 day before or 2 h after a lethal challenge of the rabies virus. Similarly, a 40 µg dose 6 h after a lethal botulinum toxin challenge entirely protected the animals. A third challenge model, where Raji-luc2 B-cell lymphoma cells were engrafted intravenously and allowed to grow for 4 days and then 10 or 50 µg of mRNA-encoded rituximab in the Acuitas LNP was administered five times over 18 days, resulted in all animals surviving this lethal tumor challenge and the 50 µg dose was able to entirely abrogate tumor growth.

Bispecific antibodies that recruit T cells to tumor cells were also encoded in modified mRNA constructs and delivered in vivo using a commercial transfection reagent, TransIT, which is not as efficient as current LNPs for liver delivery [133]. The mRNA construct could sustain circulating and bioactive bispecific antibodies for more than 6 days, while the same 5 µg dose of the protein-bispecific antibody was reduced to near baseline after one day. A second study was also carried out using bispecific antibodies in the VHH format, where one VHH that binds the conserved influenza A matrix protein 2 ectodomain (M2e) was genetically linked to a second VHH that specifically binds to the mouse Fcγ receptor IV (FcγRIV) in order to recruit innate immune cells expressing FcγRIV to influenza infected cells expressing M2e [134]. These nucleoside-modified mRNA constructs were delivered using DOTAP/cholesterol LNPs by intratracheal instillation into the mouse lung and, 4 h later, challenged with a lethal influenza virus dose. Most of the mice (80%) were protected from the lethal dose, although they did experience significant weight loss and the DOTAP/cholesterol mRNA nanoparticles resulted in a temporary influx of granulocytes in the lungs, combined with an increase in serum IL-6 cytokine levels. Finally, a potent neutralizing antibody identified in the B cells of a survivor of chikungunya infection was encoded in a nucleoside-modified mRNA construct delivered in an LNP possibly containing MC3 or Lipid 5 [135]. Protection against viral challenge administrated 24 h pre-infusion in mice was achieved at 0.5 mg/kg (10 µg) IV for the mRNA-encoded mAb, while 2 mg/kg of the protein mAb was needed. Therapeutic protection by infusion at 4 h post infection was obtained at very high doses of 10 mg/kg (200 µg) in mice. Non-human primate studies found that very high doses up to 3 mg/kg (9 mg) produced minimal transient toxicity involving splenic enlargement and increased CCL2 serum levels, and the antibody was detectable for several months post infusion. Based on these results, Moderna initiated a phase 1 human trial and announced positive results where infusions of 0.1 and 0.3 mg/kg were well tolerated and resulted in serum levels of the mAb in the 1–14 µg/mL range that are expected to be protective against chikungunya virus for up to 16 weeks after a single dose [136].

## 9. Assembly and Structure of Lipid Nanoparticles

The current methods of mRNA lipid nanoparticle production utilize microfluidic or T-junction mixing to rapidly combine an ethanol phase containing the hydrophobic lipids and an aqueous phase that contains the mRNA in a buffer, such as acetic acid, at pH 4 (Figure 2). Prior methods, such as thin film hydration and ethanol injection, are generally not used since they result in heterogeneous larger-sized nanoparticles with lower mRNA encapsulation efficiency, which are difficult to scale up [95]. Microfluidic mixing has the advantage of being able to mix very small volumes of lipids in ethanol with mRNA in aqueous solutions (tens of µL) so that the screening of many components and formulation parameters is possible. T-mixing, on the other hand, is the general method of choice for the commercial production of large batches of mRNA LNPs, such as those in current clinical trials. A recent publication demonstrated that both methods result in LNPs of similar sizes and morphologies [96]. The rapid mixing of the two solutions is key in order to limit the resultant particle size to <100 nm, thus obviating the need for the size reduction methods (extrusion, sonication) required by other production methods [137]. The assembly and formation of the LNPs from these solutions is driven by both hydrophobic and electrostatic forces, as depicted in Figure 2. The four lipids (ionizable lipid, DSPC, cholesterol, PEG–lipid) are initially soluble in ethanol without any counterions present so that the ionizable lipid is unprotonated and electrically neutral (Figure 2A). One volume of the lipid-containing ethanol solution is typically mixed with three volumes of mRNA in a pH = 4 aqueous acetate buffer so that when the lipids contact the aqueous buffer they become insoluble in a 3:1 water/ethanol solvent and the ionizable lipid becomes protonated and positively charged, which then drives it to electrostatically bind to the negatively charged phosphate backbone of the mRNA (Figure 2B), while the lipids become insoluble, forming a lipid particle encapsulating the mRNA in a primarily aqueous suspension. A key component in this process is the PEG–lipid, since the PEG chain is hydrophilic and thereby coats the particle and also determines its final thermodynamically stable size. By changing the mole fraction of PEG, the LNP size can be predictably controlled, for example, from 100 nm at a 0.5% mole fraction to 43 nm at a 3% mole fraction of PEG–lipid [89]. A recent critically important observation was that LNP structure and size continue to evolve post-mixing when the mRNA LNP suspension is either diluted in aqueous buffer or dialyzed against an aqueous buffer to both raise the pH and eliminate ethanol [96]. The initial mixing of aqueous and lipid phases produces a pH near 5.5, protonating the ionizable lipid, which has an LNP pKa of near 6.5 and allows mRNA binding and encapsulation (Figure 2B,C). Subsequent raising of the pH by dilution, dialysis or tangential flow filtration neutralizes the ionizable lipid until it is mainly uncharged at pH 7.4 (Figure 2D). As the ionizable lipid becomes neutral, it also becomes less soluble, resulting in the formation of larger hydrophobic lipid domains that drive the fusion process of the LNPs so that their size increases and the core of the LNP becomes an amorphous electron-dense phase, mainly containing the ionizable lipid bound to the mRNA. It was estimated that as many as 36 vesicles could fuse to form just one final LNP during this process (Figure 2C,D). The fusion was demonstrated using FRET pairs and the role of the PEG–lipid was further seen to occur during this process since adding the PEG–lipid after mixing controlled the final LNP size in the same way as adding the PEG–lipid before mixing [138]. This study and another study using neutron scattering methods have also shown that DSPC forms a bilayer just underneath the peripheral PEG layer in the LNP, whose central core is primarily the ionizable lipid bound to mRNA (Figure 2D). Cholesterol is thought to be distributed throughout the LNP [89].

## 10. Determinants of Performance of mRNA Delivery Systems for Vaccines

The determinants of performance for mRNA delivery systems are multifactorial and interacting and include: (1) their potency or ability to deliver to the appropriate cell and efficiently release mRNA to the cytoplasmic translational machinery; (2) their adjuvanticity, which can boost the immune response; and (3) the minimization of any contribution to adverse events or toxicity that could arise from excessive inflammation at the injection site or systemic distribution and off-target expression.

### 10.1. Dose

The potency of mRNA delivery systems is most easily appreciated by the large range of doses that are currently being pursued in SARS-CoV-2 clinical trials, from 1 to 100 µg (Table 1). Doses in human trials are clearly grouped into the higher 30–100 µg doses for nucleoside-modified RNA (Moderna, BioNTech), lower 7.5–20 µg doses for unmodified RNA (CureVac, Translate Bio), and even lower 1–10 µg doses for self-amplifying RNA (Arcturus, Imperial College of London). Two factors are at play in determining these doses: the level of neutralizing antibody titers and T cell responses achieved versus convalescent plasma, and the frequency and severity of adverse events incurred at each dose. There appears to be a fairly narrow window of acceptance where the doses required to achieve protection are also close to generating an unacceptable frequency and severity of adverse events, as evidenced by the discontinuation of the highest doses tested in all phase 1 clinical trials. Both modified nucleoside constructs tested in the BioNTech phase 1 trials had high neutralizing titers versus convalescent plasma, while the larger construct encoding the membrane-bound full-length spike protein had a lower frequency and severity of adverse events, leading to its selection for the phase 3 study. Notably, dose is represented as mass, while the molar dose is dependent on the length of the construct and, furthermore, the amount of mRNA actually being translated is a small fraction of either, depending on the efficiency and targeting properties of the delivery system.

In animal studies of prophylactic mRNA vaccines for infectious diseases, the initial doses capable of producing neutralizing antibodies or protection against viral challenge were quite high in the 10–80 µg range for mice when using protamine, dendrimers and early cationic lipid systems (Table 3). When the more recent LNPs were subsequently used, the dose required for neutralization in mice was considerably reduced to near the 1 µg level when given twice, while for non-modified mRNA the dose appears to be lower, near 0.25 µg. The dose can be lower again for self-amplifying mRNA, such as 0.1 µg given twice or 2 µg given once. In larger animal models (hamster, ferret and non-human primate), fewer studies are available and the doses fall into a wide range of 5 µg to 200 µg with no apparent pattern. Interestingly, when using body surface area to convert human doses to animal doses, a 100 µg dose for a 60 kg human would be equivalent to a 15 µg dose in a 3 kg rhesus macaque and to a 0.4 µg dose in a 20 g mouse [139], numbers that approximate those of LNPs in Table 1 and Table 3. The delivery system clearly plays an important role in determining the effective dose. There is a strong desire to improve delivery efficiency in order to reduce dose and maintain potency since this is expected to reduce adverse event frequency and severity by reducing the local reactions and off-target effects of the mRNA and of the delivery vehicle. Reducing the dose will also lower the amount of raw material needed and the cost associated with vaccinating each individual. In particular, the current COVID-19 pandemic has brought into focus some significant supply chain and production capacity limitations of mRNA LNP vaccines that could be improved with more efficient delivery systems.

### 10.2. Potency and Delivery Efficiency

There have been many studies that have attempted to identify structure–function relationships for LNP and other nucleic acid delivery systems. The most commonly cited feature of the LNP that determines its potency or delivery efficiency is its pKa. The pKa is the pH at which 50% of the ionizable lipid in the LNP is protonated. To date, the LNP pKa has only been measured with a dye-binding assay called TNS, which is negatively charged and experiences fluorescence enhancement upon binding a positively charged LNP [88]. Fluorescence measurement of LNPs incubated with TNS in buffers covering a wide range of pH values is used to deduce dye binding to surface charge and the pKa estimated, where half of the maximal fluorescence is attained. It was well established that the MC3-based Onpattro LNP had an optimal pKa of 6.4 for silencing hepatocytes after IV administration [92]. There was a very sharp optimum in TNS pKa in the range of 6.2–6.8 for any LNP to effect hepatocyte silencing. An excellent model for explaining this pKa dependence was based on the ionizable lipid in the LNP being near neutral at pH 7.4 while, after internalization into a cell, the pH of the endosome will begin to drop as it evolves through the endolysosomal pathway, thereby progressively protonating the ionizable lipid, which will then bind to an anionic endogenous phospholipid of the endosome and disrupt its bilayer structure to release the mRNA into the cytoplasm for ribosomal loading [17]. Endosomal disruption requires an additional feature of the ionizable lipid, namely a cone-shaped morphology where the cross-section of the lipid tails is larger than that of its head group. This renders the ionizable lipid/endosomal phospholipid ion pair incompatible with a bilayer and more likely to form structures such as inverted hexagonal phases that can disrupt the endosomal membrane. This has been called the molecular shape hypothesis [91] and is the mechanism explaining why the introduction of one or two double bonds into a saturated C18 alkyl chain generates a more cone-shaped and less cylindrical morphology that is membrane disrupting and endosomolytic [88]. These two C18 linoleic acid tails, combined with an appropriately tuned pKa of the dimethylamine headgroup, are the defining features of the MC3 ionizable lipid. The ionizable lipids that have replaced MC3 for mRNA delivery conserve the pKa requirement, but pursue greater endosomolytic character by introducing more branching into the alkyl tails. Lipid H and Lipid 5 from Moderna, for example, have three alkyl tails, as does Lipid 2,2 (8,8) 4C CH3 from Arcturus, while Acuitas ALC-0315 has four and A9 has five alkyl tails (Table 2). This augmented cone-shaped morphology is presumably the reason why LNPs that incorporate these ionizable lipids are more efficient delivery vehicles with greater endosomal release.

Although LNP pKa and the molecular shape hypothesis are well established as contributing to LNP delivery efficiency, other factors are important as well, such as the stability of the PEG–lipid on the LNP surface, and the proportions of the four lipids in the ethanol solution, which ultimately determine the LNP ultrastructure. The PEG–lipid controls LNP size, as mentioned above, by providing a hydrophilic shell that limits vesicle fusion during assembly so that higher PEG–lipid concentrations produce smaller LNPs. For example, one study showed that varying the mole fraction of the PEG–lipid from 0.25% to 5% reduced the LNP size from 117 nm to 25 nm and that the optimal size for hepatocyte silencing was 78 nm, generated with 2.5% PEG–lipid [142]. Since the alkyl tail of the PEG–lipid had 14 carbons, it was not stably anchored to the LNP surface and was found to be gradually shed from the LNP in circulation, along with the shedding of the ionizable lipid MC3 and DSPC. This PEG shedding is thought to render the LNP transfection competent at some point, but, if too extreme, results in the rapid loss of the ionizable lipid and DSPC, which will negatively impact endosomal release. For example, by extending the alkyl tail to 18 carbons, the PEG–lipid did not shed, but was also not silenced in hepatocytes. On the other hand, adding higher concentrations of PEG to make smaller particles resulted in faster shedding, loss of the ionizable lipid and reduced silencing. The labile and dynamic nature of the LNP is currently only partly understood. Another study also found that an intermediate sized 64 nm diameter LNP made with 1.5% PEG–lipid was more efficient for mRNA delivery than a larger one at 100 nm (0.5% PEG–lipid), as well as a smaller LNP at 48 nm (3% PEG–lipid), similar to the study mentioned above [89]. However, by changing the mole ratios of the four lipids in order to conserve a calculated density of DSPC under the PEG layer of the LNP at the optimal value found in the 64 nm 1.5% PEG–lipid LNP, these authors were able to make larger 100 nm LNPs with a twofold increase in mRNA expression compared to the 64 nm-sized LNPs. Thus, in addition to the LNP pKa, ionizable lipid molecular shape and the dynamics of the PEG–lipid, more detailed features of the LNP ultrastructure and the state of each component are also important in determining potency.

### 10.3. Endosomal Release

Cell uptake and endosomal trafficking of siRNA-LNPs were studied in detail and are assumed to be similar to the uptake and endosomal trafficking of mRNA LNPs. With the MC3 LNP, a quantitative study using colloidal gold particle counting in electron microscopy showed that only 2% of siRNA that were in endosomes actually escaped from endosomes into the cytosol, resulting in a few thousand siRNA molecules per cell that were available for silencing [106]. This number was, however, in the same range as the estimated levels of functionally active siRNAs interacting with RISC per cell at therapeutically relevant concentrations. Thus, the vast majority of siRNA was destined for lysosomal degradation or recycling through multivesicular bodies (late endosomes) for release in the exosomes [143,144]. Increasing the endosomolytic behavior of LNPs is the central approach to improving delivery efficiency, mainly through pKa adjustment of the LNP and by increasing the cone-shaped morphology of the ionizable lipid. For the latter, Lipid H [42] and Lipid 5 [101], which contain three branches versus two in MC3, but with similar pKa, increased endosomal release fourfold compared to MC3. Endosomal release has not been reported for Acuitas ALC-0315; however, its hepatocyte silencing efficiency was 10-fold higher than MC3 [47], suggesting its more cone-shaped four-branch structure also had higher endosomal release. These newer generation ionizable lipids therefore appear to achieve a endosomal release, closer to 15% or higher compared to the 2–5% found for MC3 siRNA-LNPs. One of the challenges in this area is the lack of a reliable standardized endosomal release method that can be implemented broadly. Many methods have been developed, but are usually specific to only one lab group [42,101,145,146,147,148,149]. mRNA was also recently shown to undergo exocytosis in an amount that is similar to the amount released into the cytosol [150]. MC3 LNPs disassembled in late endosomes and NP 1 complexes of MC3 and the mRNA were repackaged into exosomes that were exported from the cell. These endo–exosomes maintained an mRNA delivery capacity that was similar to the original MC3 LNPs from which they were derived, but could traffic to different tissues and appeared to be less immune activating. The potential significance of this exosomal redistribution of mRNA delivered by LNPs remains to be explored.

### 10.4. Charge and Ligand Mediated Targeting

The early lipid nanoparticles using permanently charged cationic nonionizable lipids were large and, due to their permanent positive charge, were quickly opsonized and generally targeted the lung. The group at BioNTech reduced the amount of cationic DOTMA in DOTMA/DOPE mRNA LNPs until the net charge was negative due to an excess of anionic mRNA at NP ratios of less than one. Injecting these negatively charged and large 300 nm mRNA LNPs intravenously led to spleen targeting and mRNA expression in dendritic cells and they were able to mediate adaptive as well as type I IFN-mediated innate immune mechanisms for cancer immunotherapy [151]. Similarly, spleen-targeting mRNA LNPs were produced using the C12-200 prototype LNP, but replacing C12-200 with the small dendritic ionizable lipid Cf-Deg-Lin, which has four linoleic acid alkyl chains and four nitrogen atoms with a TNS pKa of 5.7. This very low pKa of the LNP would ensure that the ionizable lipid was not protonated until it reached a pH below 7, creating an LNP that would bear a net negative charge from the mRNA until quite late in the endosomal pathway and therefore similarly traffic to the spleen [152]. They found that the major cell population in the spleen to express the mRNA were B lymphocytes, where 7% of B lymphocytes were expressed the mRNA according to flow cytometric analyses. More recently, charge-mediated targeting was achieved using three different basic LNPs with MC3, C12-200, or 5A2-SC8 as ionizable lipids mixed in a certain mole fraction of a permanently cationic lipid (DOTAP) or a permanently anionic lipid (18PA) to endow the LNPs with a net positive, net negative or an intermediate near-neutral net charge [153]. Consistent with the above findings, highly positive LNPs targeted the lungs and highly negative LNPs targeted the spleen, while intermediate charge levels predominantly targeted the liver. Liver targeting has been shown to depend on Apo-E binding to near-neutral liposomes or LNPs [154], which does not occur for negatively charged liposomes [155].

Notably, all of the above charge-mediated targeting studies have been done using IV administration and the routes typically used for vaccination, such as the intramuscular or intradermal routes, have not been examined. Most studies that analyze expression after intramuscular injection do, however, detect the systemic trafficking of mRNA LNPs, which are rapidly and strongly expressed in the liver, at the same time as they are expressed in muscle and draining lymph nodes [97,156,157]. These particular LNPs therefore seem to enter the vasculature and are subsequently expressed in liver hepatocytes due to passive ApoE-mediated targeting, which is not surprising since they were designed for hepatocyte targeting. This systemic distribution and expression of immunogens could, however, generate systemic cytokines, complement activation and lead to other potential undesirable effects that could amplify the frequency or severity of adverse events and/or impair immune response generation. Finally, only a limited number of studies have been carried out with ligand-mediated targeting of LNPs. Lung endothelial cell targeting was achieved by conjugating CD31 (PECAM) antibodies to the LNP and injecting intravascularly [158]. The liver hepatocyte-directed LNP then became largely redirected to the lung. A similar approach using a VCAM ligand successfully targeted LNPs to inflamed regions of the brain and alleviated TNF-α-induced brain edema [159]. Dendritic cells in vitro were also more efficiently transfected using a mannosylated liposome, which may be a strategy applicable to vaccination [160]. Higher throughput screening methods to identify ligands targeting specific cell types have also been developed and may be applicable for the targeting of specific dendritic cell subsets [161,162].

### 10.5. Adjuvanticity of the Lipid Nanoparticle

The lipid nanoparticle is known to have its own adjuvant activity. A study in mice at a 10 µg dose and nonhuman primates at a 100 µg dose of nucleoside-modified mRNA LNPs (from Acuitas) encoding various immunogens showed increased numbers of antigen-specific T follicular helper (Tfh) cells and germinal center B (GC B) cells compared to an inactivated virus [33]. Tfh cells drive immunoglobulin class switch, affinity maturation, and long-term B cell memory and plasma cells. An adjuvant property of the LNP itself was found when an FLuc mRNA LNP was co-administered with a protein subunit HA immunogen and increased germinal center B cell numbers fourfold, although the number of Tfh cells was not increased compared to the protein alone. The LNP thus appears to be amplifying GC B cell responses, in particular to a nucleoside-modified mRNA LNP. Another study using an asymmetric ionizable lipid from Merck investigated the use of LNPs as adjuvants for Hepatitis B protein subunit vaccines [163]. Co-administering LNPs with the protein subunit vaccine enhanced B cell responses to levels comparable to known vaccine adjuvants, including aluminum-based adjuvant, an oligonucleotide and a TLR4 agonist, 3-O-deactytaledmonophosphoryl lipid A (MPL). The LNPs elicited potent antigen-specific CD4+ and CD8+ T cell responses and the Th1 vs. Th2 bias could be further influenced by the inclusion of additional adjuvants within the LNP. A follow-up study by this group using a Dengue virus immunogen found a similarly strong adjuvant activity in the LNP and that this activity depended on the presence of the ionizable lipid [164]. The lipid components in liposomes have also been previously recognized as having adjuvant activity in mucosal vaccines [165,166].

### 10.6. Injection Site Reactions, Safety, Tolerability, Reactogenicity of mRNA LNPs

A general safety study for MC3 nucleoside-modified mRNA LNPs expressing hEPO via IV administration to liver in rats and non-human primates found mild toxicological events up to 0.3 mg/kg, which is more than 10-fold the expected therapeutic dose [167]. The main findings in the rats were increased white blood cell counts, changes in the coagulation parameters at all doses, as well as liver injury. Non-human primates showed lymphocyte depletion accompanied by mild and reversible complement activation. These results were in line with an earlier toxicological study of the same LNPs for siRNA delivery [168], where rat mortality was noted at 6 mg/kg, while the no observable adverse effect level (NOAEL) was determined to be 1 mg/kg. Above 3 mg/kg elevations to serum chemistry parameters (ALT, AST, and TBIL), hematuria, and microscopic findings in the liver (vacuolation, inflammatory cell infiltrate, fibrosis, hemorrhage, and hepatocellular necrosis), spleen (lymphoid atrophy and necrosis) and kidney (tubular degeneration/regeneration) were noted. Safety findings in patients included infusion-related reactions (15% of patients, presumably complement mediated) and transient elevations of pro-inflammatory cytokines. Notably, the above doses administered IV, such as 0.3 mg/kg, are more than 10-fold higher than those in the current SARS-CoV-2 clinical trials that use IM administration. Nonetheless, these lower doses in the current human trials still induce a high frequency and sometimes moderate severity of both local injection site reactions and systemic adverse events. Currently, there is a paucity of published animal studies regarding correlates of these human adverse events in animals.

An extensive rhesus macaque study looking at the injection sites and trafficking of mRNA expression was performed using the MC3 LNP, delivering a nucleoside-modified mRNA encoding the influenza immunogen H10 mRNA intramuscularly or intradermally at a 50 µg dose [98]. They found a rapid cell infiltrate to the injection site within 4–24 h that could be driven by the LNP alone and was mainly composed of neutrophils and monocytes. The main cell types expressing mRNA were multiple monocyte and dendritic cell subsets at the injection sites and in the draining lymph nodes. Priming of T cell responses was restricted to the draining lymph nodes and the LNP alone did not induce CD80 in antigen-presenting cells. Ongoing generation of vaccine-specific CD4+ T cells occurred only in the vaccine-draining lymph nodes, where detection of mRNA-encoded antigens peaked at 24 h, whereas the antibody responses were sustained for weeks. Results consistent with the above were also reported using a non-modified mRNA encoding rabies virus glycoprotein G (RABV-G), delivered in an Acuitas LNP to mice with 0.5–10 µg doses and to non-human primates at 10 µg and 100 µg doses [69]. They also found that the LNP alone mediated cytokine generation in the muscle injection site and draining lymph nodes, but recognized that systemic detection of IL6 could occur due to trafficking through the blood and expression in the liver. Injection site erythema and edema were noted in the non-human primates at both 10 µg and 100 µg doses. It is also interesting to note that the LNPs used in mRNA delivery systems have a size with the range of 10–100 nm, which is known to be optimal for uptake into lymphatics, and that pegylation of lipids improves retention in lymphatics [169] and can reduce complement activation [109]. Since the emergency use approval of the Pfizer/BioNTech vaccine, there has been several observed incidences of acute anaphylaxis corresponding to 1 case in 100,000 vaccinations, which is about 10-fold the rate seen with other vaccines [170]. One possible source of this anaphylaxis is the prevalence of anti-PEG antibodies in the general population, which could trigger anaphylaxis in a patient subset due to the use of the PEG–lipid in LNPs. PEG-mediated anaphylaxis has been noted, for example, in a clinical contrast agent [171] and in a liposomal formulation of doxorubicin [172]. Nonetheless, the doses administered for the current SARS-CoV-2 vaccines correspond to a total PEG dose that is at least 15-fold lower than that found in those products, which seems to diminish this possibility. Another possibility is that the reactions are anaphylactoid in nature, but are non-specific responses to inflammation and other factors. A clinical study is underway to further elucidate this issue [173].

## 11. Conclusions

The progress of mRNA therapeutics has been extraordinary over the past two decades, beginning with the identification of means to control mRNA innate immunogenicity using modified nucleosides and sequence engineering, and the application of mRNA in vaccines and other therapeutic indications. The adoption of the lipid nanoparticle prototype from that used in siRNA delivery led to an order of magnitude improvement in delivery efficiency compared to previous systems and is continually improving, mainly due to the design of new classes of ionizable lipids. Many aspects of mRNA LNP structure, function, potency, targeting and biological features, such as adjuvanticity, remain to be explored in order to fully exploit the potential of this powerful and transformative therapeutic modality.

## Figures and Tables

**Figure 1 vaccines-09-00065-f001:**
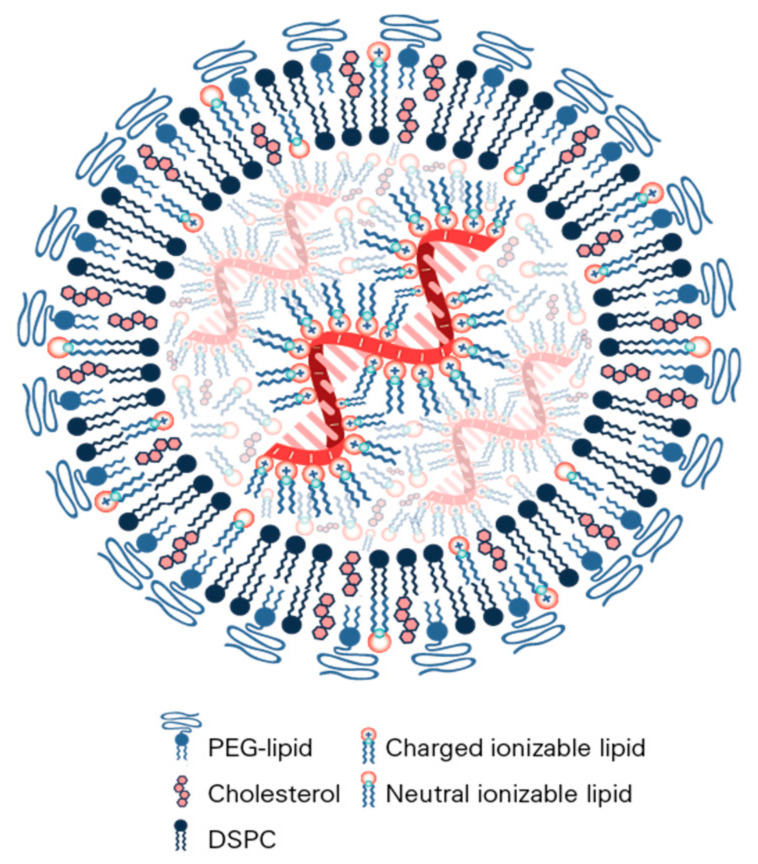
mRNA lipid nanoparticle structure. Recent studies using cryoelectron microscopy [96], small-angle neutron scattering and small-angle X-ray scattering [89] have shown that the mRNA Lipid nanoparticle includes low copy numbers of mRNA (1–10) and that the mRNA is bound by the ionizable lipid that occupies the central core of the LNP. The polyethylene glycol (PEG) lipid forms the surface of the lipid nanoparticle (LNP), along with DSPC, which is bilayer forming. Cholesterol and the ionizable lipid in charged and uncharged forms can be distributed throughout the LNP. Structural schematics of other delivery systems are available in a recent review [14].

**Figure 2 vaccines-09-00065-f002:**
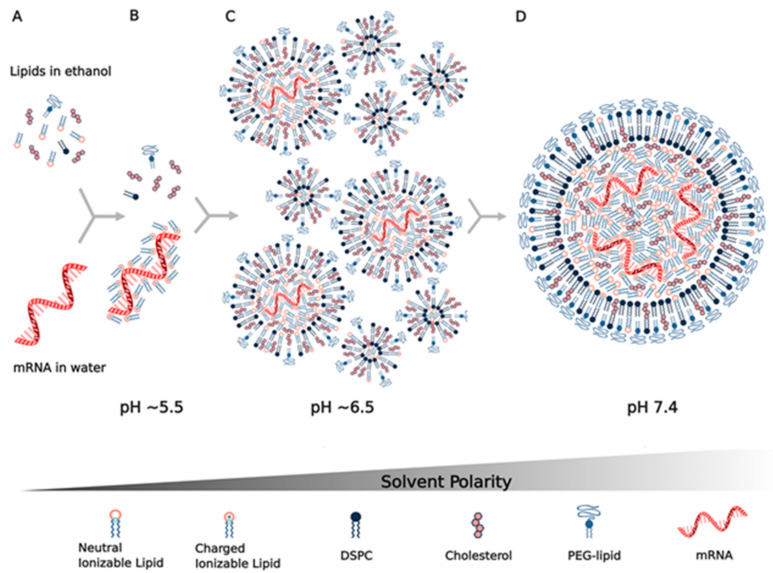
mRNA lipid nanoparticle assembly is achieved by (**A**) rapid mixing in a microfluidic or T-junction mixer of four lipids (ionizable lipid, DSPC, cholesterol, PEG–lipid) in ethanol with mRNA in an aqueous buffer near pH4. (**B**) When the ionizable lipid meets the aqueous phase, it becomes protonated at a pH ~5.5, which is intermediate between the pKa of the buffer and that of the ionizable lipid. (**C**) The ionizable lipid then electrostatically binds the anionic phosphate backbone of the mRNA while it experiences hydrophobicity in the aqueous phase, driving vesicle formation and mRNA encapsulation. (**D**) After initial vesicle formation, the pH is raised by dilution, dialysis or filtration, which results in the neutralization of the ionizable lipid, rendering it more hydrophobic and thereby driving vesicles to fuse and causing the further sequestration of the ionizable lipid with mRNA into the interior of the solid lipid nanoparticles. The PEG–lipid content stops the fusion process by providing the LNP with a hydrophilic exterior, determining its thermodynamically stable size, and the bilayer forming DSPC is present just underneath this PEG–lipid layer.

**Table 1 vaccines-09-00065-t001:** Current human trials for SARS-CoV-2 using mRNA lipid nanoparticles. All mRNA vaccines in SARS-CoV-2 clinical trials use a lipid nanoparticle for delivery. The identity and composition of each has not been publicly disclosed, so their probable class (shown below) is based on the available literature and patent citations.

Company	mRNA Type	Immunogen	mRNA Dose (µg)	Confirmed or Probable LNP Class	Publications
Moderna	nucleoside modified mRNA	membrane bound prefusion stabilized spike	100	Lipid H [42] confirmed in [41]	[43,44,45,46]
BioNTech Pfizer	nucleoside modified mRNA	membrane bound prefusion stabilized spike	30	Acuitas ALC-0315 [47] confirmed in [40]	[48,49,50,51]
CureVac	unmodified mRNA	membrane bound prefusion stabilized spike	12	Acuitas ALC-0315 [47]	[52,53]
TranslateBio Sanofi	unmodified mRNA	prefusion stabilized double mutant spike	7.5	ICE [54] or Cysteine [55]	[56]
Arcturus	self-amplifying mRNA	full length spike	1–10	Lipid 2,2 (8,8) 4C CH_3_ [57]	[58]
Imperial College	self-amplifying mRNA	membrane bound prefusion stabilized spike	1–10	Acuitas A9 [59]	[60]
Chulalongkorn	nucleoside modified mRNA	secreted wild type spike	Not available	Genevant CL1 [61]	NA

**Table 2 vaccines-09-00065-t002:** Ionizable lipids used in lipid nanoparticles. A key feature of the ionizable lipids used in lipid nanoparticles is that the pKa of the ionizable lipid in the LNP, as measured by the TNS dye-binding assay, should be in the range of 6–7. The theoretically calculated pKa of most of the ionizable groups is in the range of 8–9.5, as shown below on the nitrogen atoms, using commercial software that theoretically estimates these values in aqueous media. The 2–3 point drop in pKa from the theoretical value to the TNS value is due to the much higher energy of solvation of protons in the lipid phase, creating a pH increase of 2–3 points in the lipid compared to the aqueous phase, where pH is measured during the TNS assay [102].

Name	Ionizable Lipid Structure and Theoretical pKas	TNS pKa
MC3 [92]	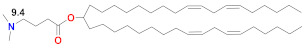	6.4
Lipid 319 [68]	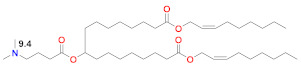	6.38
C12-200 [103]	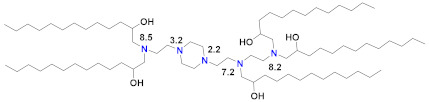	6.96
5A2-SC8 [104]	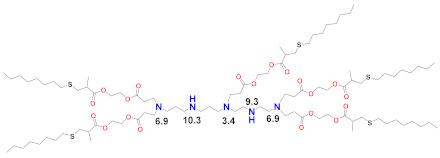	6.67
306Oi10 [105]	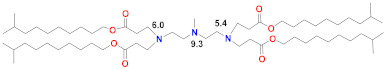	6.4
Moderna Lipid 5 [101]	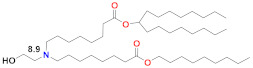	6.56
Moderna Lipid H, SM-102 [42]	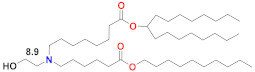	6.75
Acuitas A9 [59]	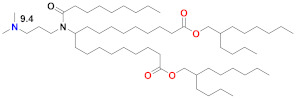	6.27
AcuitasALC-0315 [47]	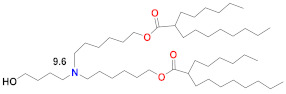	6.09
Arcturus Lipid 2,2 (8,8) 4C CH3 [57]	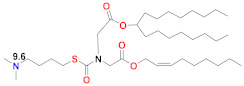	6.69
Genevant CL1[61]	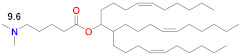	NA

**Table 3 vaccines-09-00065-t003:** mRNA doses in in vivo prophylactic vaccination. The mRNA dose required to induce neutralizing antibody titers, or the dose that provides protection against viral challenge, is shown for different mRNA delivery systems and in different species. The advent of lipid nanoparticles (LNPs) for mRNA delivery reduced the required doses by ~10-fold compared to earlier delivery systems.

Delivery System	mRNA Type	Species	Dose	Readout	Reference
Naked mRNA	non-modified	mouse	80 µg twice	protection	[140]
Naked mRNA	self-amplifying	mouse	1.25 µg twice	protection	[140]
Protamine	non-modified	mouse	10 µg twice	neutralizing titers	[66]
Protamine	non-modified	mouse	80 µg twice	protection	[66]
Modified Dendrimer	self-amplifying	mouse	40 µg once or 4 µg twice	neutralizing titers	[126]
DOTAP/DOPE/PEG	nucleoside modified	mouse	12 µg twice intranasal	neutralizing titers and protection	[130]
Cationic Nanoemulsion	self-amplifying	mouse	15 µg twice	neutralizing titers	[70]
Nanostructured Lipid Carrier	self-amplifying	mouse	0.1 µg once	neutralizing titers	[71]
LNP (Acuitas)	non-modified	mouse	0.5 µg twice	neutralizing titer	[69]
LNP (MC3)	nucleoside-modified	mouse	10 µg once or 2 µg twice	protection	[99]
LNP (MC3)	nucleoside-modified	mouse	0.4 µg once	protection	[97]
LNP (Acuitas)	nucleoside-modified	mouse	0.5 µg once	protection	[141]
LNP (Acuitas)	nucleoside-modified	mouse	1 µg twice	neutralizing titers	[49]
LNP (Moderna)	nucleoside-modified	mouse	1 µg twice	neutralizing titers and protection	[45]
LNP (Acuitas)	non-modified	mouse	0.25 µg twice	neutralizing titers	[53]
LNP (Translate Bio)	non-modified	mouse	0.2 µg twice	neutralizing titers	[56]
LNP (Arcturus)	self-amplifying	mouse	2 µg once	neutralizing titers and protection	[58]
LNP (Acuitas)	self-amplifying	mouse	0.1 µg twice	neutralizing titers	[60]
LNP (Acuitas)	non-modified	Syrian Hamster	10 µg twice	protection	[53]
LNP (MC3)	nucleoside-modified	ferret	50 µg once	neutralizing titers	[97]
Cationic Nanoemulsion	self-amplifying	non-human primate	75 µg twice	neutralizing titers	[70]
LNP (MC3)	nucleoside-modified	non-human primate	200 µg twice	neutralizing titers	[97]
LNP (MC3 or Moderna Lipid H)	nucleoside-modified	non-human primate	5 µg twice	neutralizing titers	[42]
LNP (Acuitas)	nucleoside-modified	non-human primate	30 µg twice	neutralizing titers	[49]
LNP (Moderna)	nucleoside-modified	non-human primate	100 µg twice	neutralizing titers	[45]
LNP (Translate Bio)	non-modified	non-human primate	15 µg twice	neutralizing titers	[56]

## Data Availability

No new data was generated.

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
