# Peer review of "Nanomaterial Delivery Systems for mRNA Vaccines"

_vaccines, 2021, doi:10.3390/vaccines9010065_

Round 1

Reviewer 1 Report

Brief Summary

The manuscript of Buschmann et all is a comprehensive review of delivery systems for mRNA vaccines. The available delivery systems are summarized, followed by a more detailed discussion on the development, production, delivery, efficacy  and determinants of performance of LNPs with an emphasis  on SARS-CoV-2 vaccine candidates.

Broad comments

The manuscript is very well written and the studies reviewed are well described.  I enjoyed reading the manuscript. However, I found the structure of the manuscript difficult, and it was sometimes hard to follow the story line. For example,

  • after giving information of the different delivery systems in part 2-4, part 4 and 5 go in dept on about SARS-CoV-2 LNPs, and then the story continues to general delivery systems.
  • Part 6 and 8 contain little to no research on SARS-CoV-2 LNPs, but are in between parts which focus a lot more on SARS-CoV-2 LPNs.
  • Part 7 describes research done on intranasal delivery of mRNA LNPs, but no part summarizes the other routes of administration.
  • Part 9, assembly and structure of mRNA LNPs, could be discussed earlier in the text, right after LNPs are introduced. Likely at least before part 7, in which you describe the intranasal delivery of LNPs.

To improve the readability and structure of the manuscript I would have some suggestions to consider:

  • Discuss part 6 (lipidoid) right after part 3 (polymers), and then start discussing part 4 (LNPs).
  • Add to figure 1 the structures of the other delivery systems discussed for a nicer overview.
  • Part 7: add data on other routes of administration and use subheaders, or make it more clear why intranasal requires a whole paragraph and other routes of administration are not discussed.

Part 8: I do not fully understand why the manuscript contains this part about the delivery of mRNA LNPs encoding antibodies. While this is a very interesting technology, little here is explained on the delivery of these LNP, and most of the text is explaining the results of the described studies. Are the delivery systems used for these antibody LNPs different than the LNPs already described in the manuscript? If yes, this should be mentioned more cleary. If this is not the case, I am open to the idea of removing this paragraph from the manuscript.

Specific comments

Line 155: Add “-“ between IL (Interleukin) and number (eg. 6). This is done inconsistently.

Table 2: The legend is used to explain the methodology to test the pKa and the possible ranges of this value. I would suggest to move this to the main text, and leave the legend to solely describe what the table is showing. Additionally, some pKa values are covered by the image of the molecule structure, this should be corrected.

Part 5: I am open to the idea to use subheader here, to separate the different companies and their LNPs.

Figure 2: move the pH text up to ensure the reader does not think the pH is part of the solvent polarity bar.

Author Response

We thank the reviewer for his/her helpful and detailed comments that have improved the manuscript. Specific responses to the reviewer's request follows : 

Reviewer Comment : Discuss part 6 (lipidoid) right after part 3 (polymers), and then start discussing part 4 (LNPs).

Response : We do not recommend this change since lipidoids need to follow section 4 that introduces the Onpattro prototype that many lipidoid formulations are based on.

Reviewer Comment : Add to figure 1 the structures of the other delivery systems discussed for a nicer overview.

Response : This review focuses on lipid nanoparticles since the other delivery systems have been reviewed extensively in recent reviews and have not evolved to a level that is competitive with lipid nanoparticles. There is not much new to report on for other systems since the prior reviews in 2019 and 2020 unlike for Lipid Nanoparticles where much development in the past year has occurred. For these reasons we feel it is sufficient to refer to published recent reviews for the other delivery systems. We added to the figure caption a reference that provides the other requested schematics “Structural schematics of other delivery systems are available in a recent review” citing Pardi 2018.

Reviewer Comment : Part 7: add data on other routes of administration and use subheaders, or make it more clear why intranasal requires a whole paragraph and other routes of administration are not discussed.

Response : The other routes of administration are covered throughout the rest of the review, mainly intramuscular but also intradermal so that they do not require their own section. Intranasal is less developed but promising and important so it needed its own section. We clarified this at the beginning of the intranasal section adding “For mRNA vaccines the vast majority of studies and all current clinical trials have used intramuscular administration while intradermal administration has also been studied, usually in parallel with the intramuscular route. Although not highly developed to date, intranasal administration of vaccines presents the advantages of activating mucosal immunity that is very relevant for respiratory pathogens and reducing reliance on needle-based immunizations.”

Reviewer Comment : Part 8: I do not fully understand why the manuscript contains this part about the delivery of mRNA LNPs encoding antibodies. While this is a very interesting technology, little here is explained on the delivery of these LNP, and most of the text is explaining the results of the described studies. Are the delivery systems used for these antibody LNPs different than the LNPs already described in the manuscript? If yes, this should be mentioned more cleary. If this is not the case, I am open to the idea of removing this paragraph from the manuscript.

Response : mRNA-encoded antibodies were included in their own section since they can be used to protect prophylactically against infectious disease, as in the chikungunya example provided, and in the context of mRNA therapeutics are interesting to include. The LNPs used to deliver these mRNA-encoded antibodies are cited in this section and are from Acuitas and Moderna that are described in detail prior to this section, as well as the commercial delivery systems TransIT.

Specific comments

Reviewer Comment : Line 155: Add “-“ between IL (Interleukin) and number (eg. 6). This is done inconsistently.

Response : This change has been made.

Reviewer Comment : Table 2: The legend is used to explain the methodology to test the pKa and the possible ranges of this value. I would suggest to move this to the main text, and leave the legend to solely describe what the table is showing. Additionally, some pKa values are covered by the image of the molecule structure, this should be corrected.

Response : Since there are two sets of pKa values in the table and we feel it is most clear and direct to indicate in the table legend how they were measured and why they are different. We acknowledge this to be a fairly high level of detail for the legend but it is more efficient than disrupting the flow of the text elsewhere to explain these differences. We could not see any pKa values covered by the image of the molecule structure.

Reviewer Comment : Part 5: I am open to the idea to use subheader here, to separate the different companies and their LNPs.

Response : We added the subheaders.

Reviewer Comment : Figure 2: move the pH text up to ensure the reader does not think the pH is part of the solvent polarity bar.

Response : This change has been made.

Reviewer 2 Report

The manuscript by Buschmann et al., is  well written review that summarizes the technology involved in nanodelivery system of mRNA based vaccines and the clinical outcomes of the current SARS-CoV-2 vaccine trails. The major and minor suggestions are:

Major: The authors need to add a section on the challenges with distribution of mRNA vaccines.

Minor: The authors need to correct MRNA to mRNA in all the citations included in the references.

Author Response

We thank the reviewer for his/her helpful and detailed comments that have improved the manuscript. Specific responses to the reviewer's request follows : 

Reviewer Comment : Major: The authors need to add a section on the challenges with distribution of mRNA vaccines.

Response : We have added a section i) Storage and Distribution at the end of section 5.

Reviewer Comment : Minor: The authors need to correct MRNA to mRNA in all the citations included in the references.

Response : This change has been made.